# Distribution of the Water-Soluble Astaxanthin Binding Carotenoprotein (AstaP) in Scenedesmaceae

**DOI:** 10.3390/md19060349

**Published:** 2021-06-20

**Authors:** Hiroki Toyoshima, Ami Miyata, Risako Yoshida, Taichiro Ishige, Shinichi Takaichi, Shinji Kawasaki

**Affiliations:** 1Department of Bioscience, Tokyo University of Agriculture, 1-1-1 Sakuragaoka, Setagaya-ku, Tokyo 156-8502, Japan; hiroki.toyoshima91@gmail.com (H.T.); e-perfect7-x.hbk@i.softbank.jp (A.M.); lisakoyosida@icloud.com (R.Y.); 2NODAI Genome Research Centre, Tokyo University of Agriculture, 1-1-1 Sakuragaoka, Setagaya-ku, Tokyo 156-8502, Japan; t3ishige@nodai.ac.jp; 3Department of Molecular Microbiology, Tokyo University of Agriculture, 1-1-1 Sakuragaoka, Setagaya-ku, Tokyo 156-8502, Japan; st206165@nodai.ac.jp

**Keywords:** astaxanthin, *Scenedesmus*, *Coelastrella*, carotenoprotein, astaxanthin-binding protein

## Abstract

Photooxidative stress-inducible water-soluble astaxanthin-binding proteins, designated as AstaP, were identified in two Scenedesmaceae strains, *Coelastrella astaxanthina* Ki-4 and *Scenedesmus obtusus* Oki-4N; both strains were isolated under high light conditions. These AstaPs are classified as a novel family of carotenoprotein and are useful for providing valuable astaxanthin in water-soluble form; however, the distribution of AstaP orthologs in other microalgae remains unknown. Here, we examined the distribution of AstaP orthologs in the family Scenedesmaceae with two model microalgae, *Chlamydomonas reinhardtii* and *Chlorella variabilis*. The expression of AstaP orthologs under photooxidative stress conditions was detected in cell extracts of Scenedesmaceae strains, but not in model algal strains. Aqueous orange proteins produced by Scenedesmaceae strains were shown to bind astaxanthin. The protein from *Scenedesmus costatus* SAG 46.88 was purified. It was named ScosAstaP and found to bind astaxanthin. The deduced amino acid sequence from a gene encoding ScosAstaP showed 62% identity to Ki-4 AstaP. The expression of the genes encoding AstaP orthologs was shown to be inducible under photooxidative stress conditions; however, the production amounts of AstaP orthologs were estimated to be approximately 5 to 10 times lower than that of Ki-4 and Oki-4N.

## 1. Introduction

Under the unfavorable conditions combined with high light irradiation, photosynthetic organisms feel photooxidative stress [1,2]. Plants are known to use carotenoids to dissipate excess light energy. Carotenoids are distributed in many organisms, such as chloroplasts of plants and algae, animal skins, and fish eggs, and more than 800 natural carotenoids have been identified [3]. They are hydrophobic; however, only a few are known to be water-soluble by binding proteins. Water-soluble carotenoproteins have been identified, such as astaxanthin-binding crustacyanin from crustaceans [4,5], zeaxanthin-binding GSTP1 (glutathione S-transferase like protein) from the human eye [6], lutein-binding protein from silkworm [7], cyanobacterial orange carotenoid protein (OCP) [8], and eukaryotic microalgal AstaP (astaxanthin binding protein) [9,10]. These water-soluble carotenoproteins did not show structural relationships with each other [9].

In photosynthetic organisms, water-soluble carotenoproteins (WSCPs) have been well characterized in cyanobacteria named orange carotenoid protein (OCP) [8,11,12]. In our previous study, two types of novel WSCPs were identified from the eukaryotic microalgae, *Coelastrella astaxanthina* Ki-4 and *Scenedesmus obtusus* Oki-4N [9,10]. These two strains were shown to produce novel astaxanthin-binding water-soluble proteins. Both AstaPs belonged to the fasciclin protein family. This family is characterized as secreted and cell surface proteins; however, to our knowledge, none of the proteins have been reported to bind lipids, including carotenoids. Based on the photooxidative stress inducible profiles and the potent activities toward ^1^O_2_ quenching with astaxanthin binding, these novel types of astaxanthin-binding proteins are proposed to be involved in a unique function of photooxidative stress protection in plants. In this study, we investigated the distribution and characteristics of WSCPs in taxonomically related Scenedesmaceae strains.

## 2. Results

### 2.1. Effect of Photooxidative Stresses on the Tested Strains

Scenedesmaceae strains were obtained from culture collections, as described previously [11]. *S. obtusus* and *C. striolata* were selected because they are the type species of the genera *Scenedesmus* and *Coelastrella*, respectively. Other strains were chosen as the relatively well-used research strains in the genus *Coelastrella*. According to the locality information in the literature, these strains were isolated from non-stressed environmental conditions, such as lakeside and peat bog [13]. The test strains showed good growth in the medium used for the Ki-4 and Oki-4N strains [10]; therefore, these strains were subjected to photooxidative stress after growing under non-stressed conditions.

Strains Ki-4 and Oki-4N showed the optimum expression level of AstaP when they were grown under 0.7 M and 0.5 M NaCl stress with high light exposure conditions (800 µmol photons m^−2^ s^−1^), respectively, where the color of cells changed from green to orange [9,10]. The tested Scenedesmaceae strains’ cell color turned from green to white under the same stress conditions as Ki-4; therefore, we examined the upper limits of salt concentrations for each strain under the same high light intensity. Figure 1 shows the change in cell color of the strains after one to two weeks from the start of stress treatments using the upper limit of salt concentrations for each strain under high light exposure conditions. *S. obtusus* and *S. obliquus* turned green to white under 0.3 M NaCl and 0.25 M NaCl with high light exposure conditions; therefore, these strains were stressed by 0.25 M and 0.2 M NaCl, respectively, under the high light exposure conditions. Other strains were stressed with 0.5 M NaCl (for *C. vacuolata*) or 0.4 M NaCl (for *S. costatus* and *C. striolata*) with high light exposure.

A 1.0 g aliquot of wet cells stressed by salt with a high light exposure was suspended in 9.0 mL of 50 mM Tris-HCl buffer, pH 7.5. Cells were broken by a bead beater instrument, and the aqueous supernatants were obtained after ultracentrifugation. Strain Ki-4 and Oki-4N produced orange supernatants with broad absorption peak at 484 nm (Table 1) [9,10]. Except for *S. obliquus*, the test strains produced pale orangish or yellowish supernatants. The color of aqueous supernatants from the Scenedesmaceae strains were difficult to detect; therefore, the supernatants were concentrated by using a protein concentrator, and the colors and the absorption spectrum for each strain were shown in Figure 2. The amount of pigment produced per aliquot of cells from each test strain was estimated to be less than 5–10 times lower than that of Ki-4 and Oki-4N (Table 1).

These orange supernatants were passed through gel-filtration column chromatography, and two major peaks were detected (Figure 3). The first peaks from each test strain were detected around the HPLC retention time of 18 min, which were estimated to have a molecular weight greater than 670 kDa, and the absorption peaks were observed at approximately 454, 484, and 674 nm. These data indicated that the first peaks contained chlorophyll, which resembled that of the first peak of the strain Oki-4N described in our previous study [10]. The second major peak in each strain was detected at the HPLC retention time of 28 min (*S. obtusus*) or 30–35 min, which showed a broad absorption peak at 484 nm. In our previous study, AstaP orthologs were classified into two groups: glycosylated large-sized AstaPs and non-glycosylated small-sized AstaPs. The difference in the second peak retention time indicated variations in molecular weight among the test strains.

The protein fractions from the second peak in each test strain were collected, and the binding pigments were extracted as described previously [9,10]. The elution profiles were compatible with those from the strains Ki-4 and Oki-4N (Figure 4), and the pigments from each strain were identified to be astaxanthin and adonixanthin based on C_18_-HPLC retention time and absorption spectrum. These results indicated that the test strains also produced AstaP-like water-soluble proteins under the photooxidative stress conditions.

### 2.2. Effect of Photooxidative Stress Response on Model Microalgae

In our previous study, AstaP orthologs are found in the genome annotation data of *Chlamydomonas reinhardtii* and *Chlorella variabilis* [8]. In this study, we analyzed the production of aqueous pigments from two strains under photooxidative stress conditions. *Cmy. reinhardtii* and *Crl. variabilis* turned green to white under 0.25 M and 0.5 M NaCl conditions with high light intensity, respectively. Therefore, photooxidative stresses were archived under 0.2 M and 0.4 M NaCl, respectively, with high light for six days. The cell-free extracts from photooxidative-stressed cells were obtained by a bead beater, ultracentrifuged, and their color was pale yellow. A yellow fraction was detected using gel filtration HPLC of the aqueous supernatant at 18 min in *Crl. variabilis* but not in *Cmy. reinhardtii*. The yellow fraction of *Crl. variabilis* showed absorption peaks at 454, 484, and 674 nm, indicating the presence of chlorophyll-binding protein similar to that of other Scenedesmaceae strains. These data suggested that the production of aqueous carotenoid-binding protein was not detectable in these model microalgae.

### 2.3. Purification and Characterization of Water-Soluble Astaxanthin-Binding Protein from S. costatus SAG 46.88

In order to characterize the existence of AstaP orthologs in the test strains, we chose *S. costatus* for further purification of astaxanthin binding protein because of its relatively high expression level (Table 1). The aqueous orange fraction obtained by gel-filtration column chromatography from the stressed *S. costatus* CFE was further purified by isoelectric focusing. An orange band that migrated to the position corresponding to pI 9 was detected (Figure 5A), and the purity of the excised orange protein band was determined by SDS-PAGE (Figure 5B). A single band appeared as an apparent molecular mass of 27 kDa, and the N-terminal amino acid sequence was determined to be AVPEAKTT, which showed partial homology to the N-terminal sequence of Ki-4-AstaP (ATPKANAT). The binding pigments of the purified protein by isoelectric focusing were analyzed by C_18_-HPLC and were astaxanthin and adonixanthin, which were identical to those of Ki-4 AstaP [9]; therefore, we named this protein ScosAstaP.

Two-dimensional polyacrylamide gel electrophoresis (2D PAGE) was performed to detect the photooxidative stress-inducible spots in *S. costatus*, and a large spot was detected in the area of alkaline pI after exposure to salt stress with high light (Figure 5). The N-terminal amino acid sequence (AVPEAKTT) from the large spot coincided with that of the purified protein. These results indicated that the orange protein was an AstaP ortholog inducible under the photooxidative stress conditions.

### 2.4. AstaP Ortholog in S. costatus Is a Fasciclin-Like Glycosylated Protein

To obtain the gene encoding ScosAstaP, a cDNA library was constructed from the stressed *S. costatus* mRNA. The gene encoding the N-terminal amino acid sequence was found in the de novo sequencing data of the cDNA library, and the full-length cDNA was obtained by PCR using the *S. costatus* cDNA library as a template. The deduced amino acid sequence from a gene encoding ScosAstaP showed 62% identity with Ki-4 AstaP (Figure 6A). Bioinformatics analyses revealed that ScosAstaP conserved 20 amino acid residues of the N-terminal hydrophobic signal sequence and two fasciclin-like H1 and H2 domains. These protein primary structures are similar to Ki-4-AstaP; however, only one putative N-glycosylation Asn-x-Thr site was detected in ScosAstaP compared to the five sites in Ki-4 AstaP (Figure 6A). The calculated molecular mass of the mature protein deduced from ScosAstaP cDNA, excluding 20 amino acid residues of the N-terminus signal peptide, was 21.4 kDa. The reason for the difference of calculated molecular weight and the measured molecular weight (apparent molecular weight from SDS-PAGE was 27kDa) is supposedly due to the glycosylation of ScosAstaP like as Ki-4 AstaP (Ki-4 AstaP was 21.2 kDa of the calculated molecular mass and the apparent molecular weight from SDS-PAGE was 33 kDa). Northern blot analysis indicated that the gene encoding ScosAstaP was inducible by photooxidative stress (Figure 7).

### 2.5. Identification of AstaP Orthologs in C. vacuolata and Two Model Microalgae

*C. vacuolata* showed two peaks of astaxanthin-binding protein by gel-filtration chromatography (Figure 3); therefore, the presence of two types of AstaP orthologs was predicted. Two cDNA sequences that encode AstaP orthologs were found in the cDNA sequence data from the photooxidative-stressed *C. vacuolata* cDNA library. Both full-length cDNA clones were amplified by PCR, and nucleotide sequences were confirmed. The deduced amino acid sequence indicated that *C. vacuolata* possesses two types of AstaP orthologs. These two proteins showed different pI (calculated pI were 6.7 and 9.7), and we named them CvacAstaP1 and CvacAstaP2, respectively. Both orthologs possess N-glycosylation sites and N-terminal hydrophobic signal peptides for cell surface secretion, and they showed 44% identity with each other (Appendix A). ScosAstaP1 had ten putative N-linked glycosylation sites, whereas ScosAstaP2 had only two putative N-glycosylation sites. The number of N-glycosylation sites was expected to affect the difference in molecular weight detected by gel-filtration column chromatography, as shown in Figure 3.

AstaP orthologs were found in the model algae *Cmy. reinhardtii* and *Crl. variabilis,* and phylogenetic analysis was performed in our previous study. Both cDNAs were amplified by PCR and confirmed the coincidence of the deduced amino acid sequence in the database (Appendix A). To determine the gene expression profiles of AstaP orthologs in model algae and *C. vacuolata*, the genes encoding *Cmy. reinhardtii*, *Crl. variabilis*, and two AstaP orthologs from *C. vacuolata* were amplified and analyzed these expression profiles by Northern blot analysis. Both AstaP orthologs were induced by photooxidative stress treatments.

The *Scenedesmus actus* AstaP ortholog was identified as a translation product of a gene induced by Cr stress [14], but its binding of astaxanthin has not been determined. Phylogenetic analysis revealed that the AstaP orthologs were classified into three groups: glycosylated basic pI, glycosylated acidic pI, and non-glycosylated acidic pI (Figure 8).

## 3. Discussion

In this study, we investigated the distribution of AstaP orthologs from Scenedesmaceae strains. Although AstaP proteins have not been identified except strains that were isolated under high light conditions in our previous study, new AstaP orthologs from Scenedesmaceae strains were identified in the strains from culture collections. All the AstaP orthologs commonly conserved H1 and H2 fasciclin domains and N-terminal hydrophobic signal peptide for secretion; however, the number of glycosylation sites, pI, and molecular weights were found to vary depending on the strain (Appendix A). *Cmy. reinhardtii* and *Crl. variabilis* were shown to possess photooxidative stress inducible AstaP orthologs; however, we could not detect the expression of water-soluble carotenoproteins in both strains. Further analysis will be needed to make the function of these gene products clear including their possibilities for carotenoid binding.

Scenedesmaceae strains from the culture collection were isolated under non-stressful conditions, such as in lake (*S. costatus*) [15], bark (*C. vacuolata*) [16], and peat bog (*C. striolata*) [13,17,18]. All the test strains, except *S. obliquus*, were found to express astaxanthin-binding water-soluble protein under the experimental conditions in this study; however, the amounts of production were estimated to be low compared to those of the Ki-4 and Oki-4N. We conclude that AstaP is widely distributed in Scenedesmaceae, and the expression levels of AstaP may correlate with the ability of photooxidative stress tolerance of each strain. 

In conclusion, although the functional differences among these AstaP orthologs remain unclear, the presence or absence of glycosylation, the difference in pI, and the presence or absence of signal peptide for secretion may reflect the subcellular localization of AstaPs. In our previous study, Ki-4 AstaP, which is classified as a glycosylated basic pI group, was predicted to be localized at the cell surface [9,19]. On the other hand, Oki-4N AstaP-pinks, which were classified as a non-glycosylated acidic pI group, were suggested to be localized at the endoplasmic reticulum or vacuole but not at the cell surface based on the fluorescent microscopic analysis [10]. Further characterization of AstaP orthologs will clarify the distribution, localization, and specific functions under the photooxidative stress conditions.

## 4. Materials and Methods

### 4.1. Microalgal Strains and Growth Conditions

Scenedesmaceae strains were obtained from culture collection center as previously described [13]. Strain Ki-4 and Oki-4N were isolated in our laboratory as previously described [9,10]. *Scenedesmus* sp. Oki-4N was named to *Scenedesmus obtusus* Oki-4N based on a high similarity of homology score of 18S rDNA and ITS2 sequence with the morphological identity with *Scenedesmus obtusus* as previously described [10]. The strain name *Coelastrella vacuolata* (synonym: *Chlorella fusca* var *vacuolata* = *Scenedesmus vacuolatus = Graesiella vacuolata*) [13,16,18,20] was used in this study. The composition of the modified A3 medium was the same as previously described [9,10]. Algal strains were cultivated in A3 medium with a 16h/8h light/dark regime at 26 °C under low light conditions (60 µmol photons m^−2^ s^−1^) as previously described [18]. Photooxidative stress was induced by the addition of sterilized NaCl when the growth reached an OD_750_ value of 1.0 cm^−1^ under conditions of high light exposure (~800 µmol photons m^−2^ s^−1^) as previously described [9,10]. Cells did not change color to orange under high light conditions (800 µmol photons m^−2^ s^−1^) without salt, or low light conditions with salt.

### 4.2. Gel Filtration Column Chromatography

The water-soluble astaxanthin binding protein fractions were obtained by gel filtration column chromatography as previously described. Briefly, stressed cells were harvested, and 1.0 g of stressed wet cells were suspended in 9.0 mL of 50 mM Tris-HCl buffer at pH 7.5. Cells were broken by a multi-beads shocker (Yasui Kikai, Osaka, Japan), dissolved in Tris-buffer pH 7.5. Cell extracts (CFEs) were ultracentrifuged at 100,000× *g* for 2 h to remove cell debris and lipids. The CFEs were passed through a Sephacryl S-200 HR gel filtration column at a flow rate of 1.0 mL/min (1.6 × 60 cm, GE Healthcare, Chicago, IL, USA). The elution profiles were monitored using a photodiode array detector LaChrome Elite software (Hitachi Ltd., Tokyo, Japan), and fractions of orange eluates were collected. Proteins were concentrated by using Amicon Ultra (Merck, Darmstadt, Germany).

### 4.3. Pigment Extraction and Identification

The binding pigments of water-soluble carotenoprotein were extracted using the Bligh–Dyer method as previously described [9,10,19,21]. Pigments were extracted with methanol: chloroform: H_2_O = 12:5:3 by gentle mixing in a tube. After the addition of chloroform and water (2:3), the organic phase was obtained and evaporated to dryness under N_2_ gas. The extracted pigments were completely dissolved in acetone and analyzed by C_18_-HPLC (CAPCEL PAK C18 reversed-phase column, 150 × 4.6 mm, flow rate 1.0 mL/min). The pigment was identified based on the absorption spectra obtained using an HPLC photodiode array detector, HPLC retention times, and molecular masses from high-resolution LC/MS analysis in comparison with those of standard compounds.

### 4.4. Purification of the Water Soluble Astaxanthin Binding Protein

The water-soluble astaxanthin binding protein of *S. costatus* was purified by isoelectric focusing. Isoelectric focusing was performed horizontally with Maltiphor II (GE Healthcare, Chicago, IL, USA). Migrated orange band was excised and electrophoresed by SDS-PAGE. As standard markers, Precision Plus Protein^TM^ standard kit (Bio-Rad, Berkeley, CA, USA) was used.

### 4.5. Determination of Peptide Sequence

The N-terminal amino acid sequence was determined by the Edman degradation method using a PPSQ30 peptide sequencer (Shimadzu, Tokyo, Japan) as previously described [9]. Target protein band was obtained using a transferred protein on a PVDF membrane stained with Coomassie Brilliant Blue and digested with trypsin (Promega, Madison, WI, USA).

### 4.6. Isolation of RNA and Construction of a cDNA Library

Isolation of RNA and construction of a cDNA library were performed as previously described [9,10]. To prepare a cDNA library, *S. costatus* and *C. vacuolata* cells that were subjected to NaCl stress under high light stress and total RNA were extracted with Trizol reagent (Invitrogen, Waltham, MA, USA). Poly(A)^+^ mRNA was isolated from total RNA and used to generate a full-length cDNA library with a Smart-Infusion PCR cloning system (Clontech, Palo Alto, CA, USA). The cDNA libraries were sequenced using the Illumina HiSeq 2500 system (Illumina, San Diego, CA, USA) and assembled de novo sequence by CLC Genomics Workbench (Qiagen, Hilden, Germany).

### 4.7. Bioinformatics

Bioinformatic analyses were performed as previously described [9,10]. Briefly, database searches for sequence homology were performed using the programs BLASTp (http://www.ncbi.nlm.nih.gov/BLAST/, accessed on 19 June 2021) and FASTA (http://www.genome.jp/tools/fasta/, accessed on 19 June 2021) set to standard parameters. The putative cell localization of AstaP was predicted using PSORT (http://psort.hgc.jp/form.html, accessed on 19 June 2021), SignalP (http://www.cbs.dtu.dk/services/SignalP/, accessed on 19 June 2021) and TargetP (http://www.cbs.dtu.dk/services/TargetP/, accessed on 19 June 2021). The *O*-linked glycosylation of Ser and Thr residues was predicted by the program NetOGlyc 3.1 (http://www.cbs.dtu.dk/services/NetOGlyc/, accessed on 19 June 2021). The *N*-linked glycosylation site was predicted by the program NetNGlyc 1.0 (http://www.cbs.dtu.dk/services/NetNGlyc/, accessed on 19 June 2021). The isoelectric point (pI) and molecular mass were predicted by GENETYX-MAC software (GENETYX Corporation, Tokyo, Japan).

### 4.8. Northern Hybridization

Northern hybridization was performed as previously described [9,10]. Briefly, total RNA was isolated using TRIzol (Invitrogen, Waltham, MA, USA) from the cells that were harvested at several time points after applying salt stress under high light conditions. The total RNA (10 µg) was subjected to electrophoresis in 1.0% agarose gels, blotted onto nylon Hybond N^+^ membranes (Amersham, Chicago, IL, USA) were probed with the PCR-amplified DNA fragment encoding the target region. The identity of the amplified DNA fragment was confirmed by size and nucleotide sequence. The RNA was pre-hybridized at 60 °C for 30 min, and the ^32^P-labeled DNA probe was hybridized to RNA on the membrane at 60 °C for 12 h.

### 4.9. Two-Dimensional Electrophoresis

Two-dimensional electrophoresis was performed as previously described [9,10]. Untreated *S. costatus* cells and exposed to 0.4 M NaCl stress under high light conditions were harvested and promptly frozen in liquid nitrogen. Proteins of several samples were sedimented by acetone, and the protein pellet was dried in vacuo. Dried pellet was dissolved in a two-dimensional (2D) electrophoresis sample buffer containing 8 M urea, 30 mM DTT, 2% (*v*/*v*) Pharmalyte 3–10 (Pharmacia), and 0.5% Triton X-100. Both isoelectric focusing (IEF) and SDS-PAGE were performed horizontally with Maltiphor II (Pharmacia, Peapark, NJ, USA). For electrophoresis, 40 µg of total protein was loaded onto gels. For detection, the gels were silver-stained with a Silver Staining Kit (Invitrogen).

## Figures and Tables

**Figure 1 marinedrugs-19-00349-f001:**
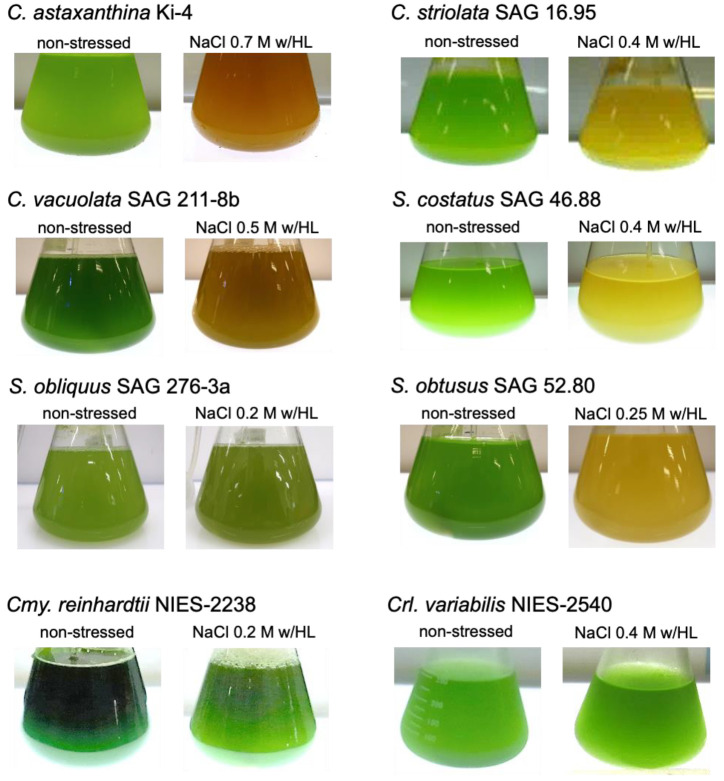
Survival of test strains under salt stress conditions with high light exposure. Salt concentrations for each test strains were decided based on the upper limit of salt concentrations with high light (w/HL, 800 µmol photons m^−2^ s^−1^). Each test strain was cultivated under non-stressed conditions and stressed by adding salt w/HL for one to two weeks as described in Materials and Methods.

**Figure 2 marinedrugs-19-00349-f002:**
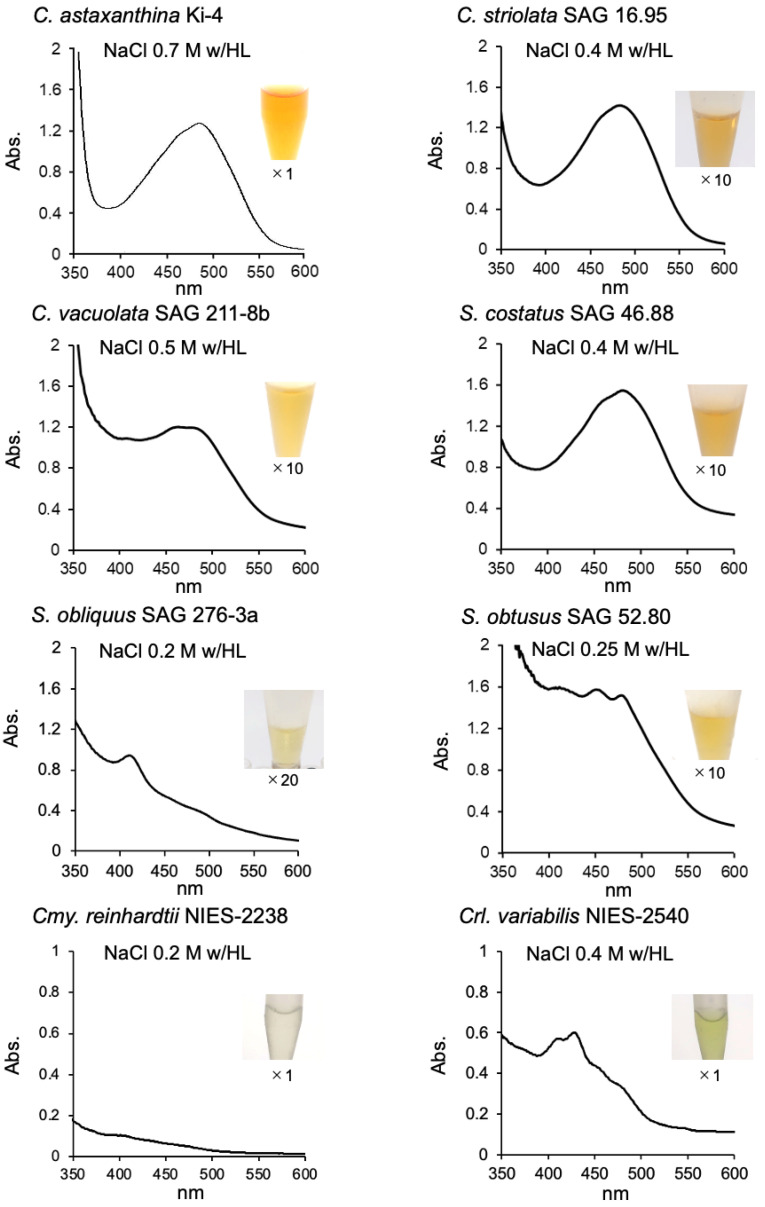
Absorption spectrum and the color of aqueous cell extracts. Aqueous cell extracts from photooxidative stressed cells were obtained after ultracentrifugation and concentrated by using a protein concentrator. ×1, ×10, ×20 means the rate of concentration by using a protein concentrator.

**Figure 3 marinedrugs-19-00349-f003:**
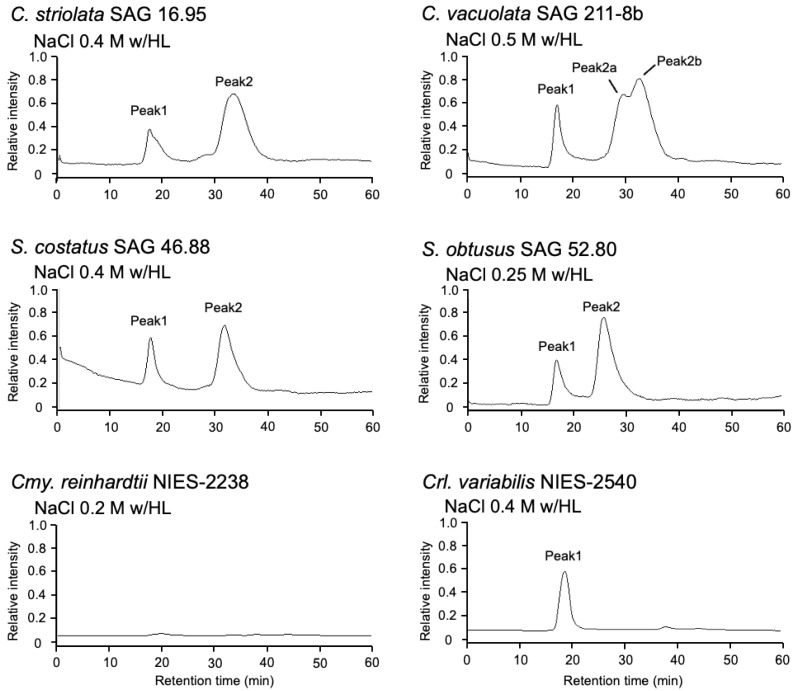
Separation of carotenoid pigments by gel-filtration column chromatography. Aqueous cell extracts obtained after ultracentrifugation were loaded into the gel-filtration column chromatography.

**Figure 4 marinedrugs-19-00349-f004:**
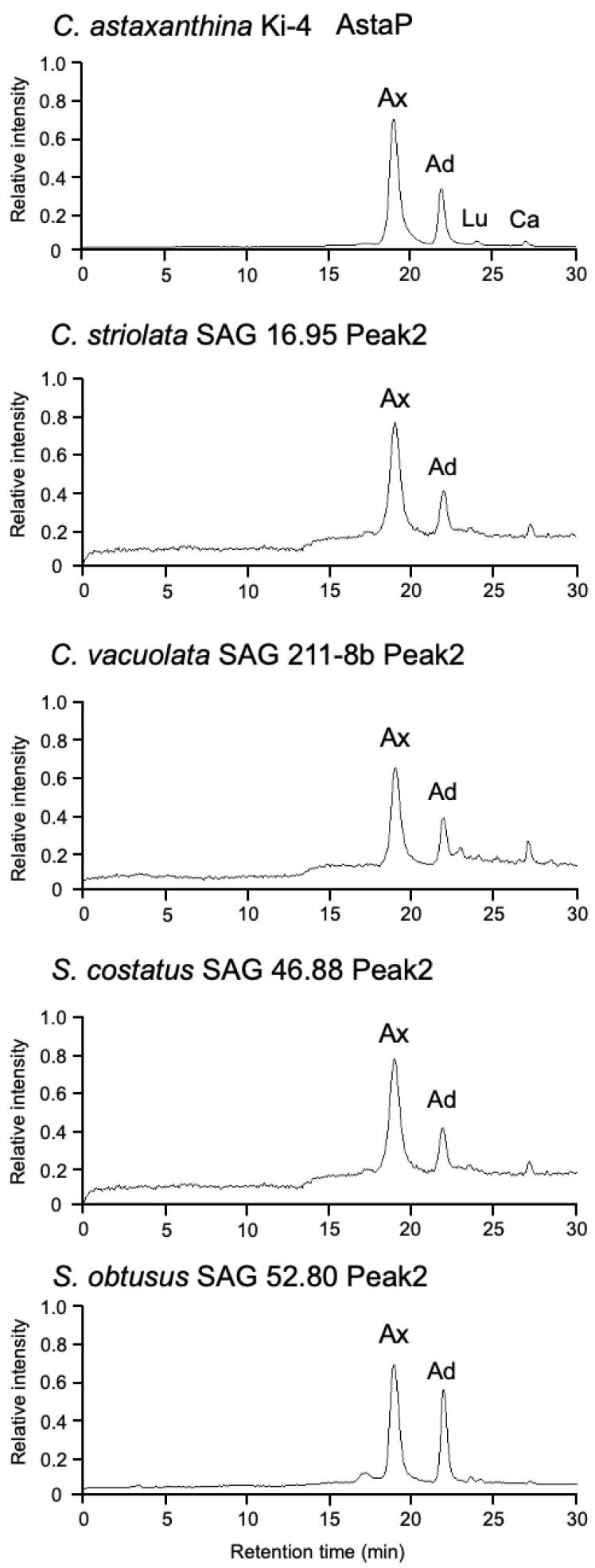
C_18_-HPLC elution profiles of the binding carotenoids from the collected Peak2 fractions after gel filtration column chromatography. Ax, astaxanthin; Ad, adonixanthin; Lt, lutein; and Ca, canthaxanthin.

**Figure 5 marinedrugs-19-00349-f005:**
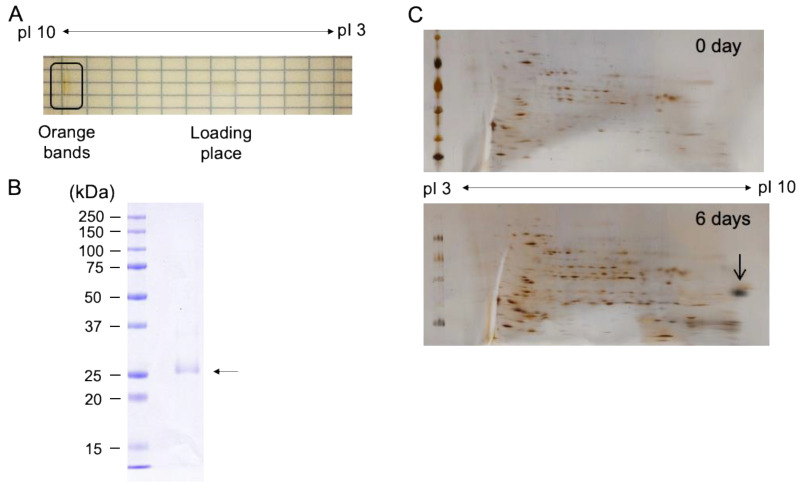
Purification and characterization of AstaP from *S. costatus*. (**A**) isoelectric focusing of aqueous pigments. The circle indicates the migrated orange bands after the electrophoresis; (**B**) the migrated orange band in A excised and electrophoresed by SDS-PAGE. The purified protein is shown by an arrow. Molecular mass standards are shown (kDa); (**C**) two-dimensional electrophoresis of the total proteins of *S. costatus*. *S. costatus* was cultivated under high light conditions and subjected to 0.4 M NaCl for six days. An arrow indicates the excised spot for N-terminal sequencing.

**Figure 6 marinedrugs-19-00349-f006:**
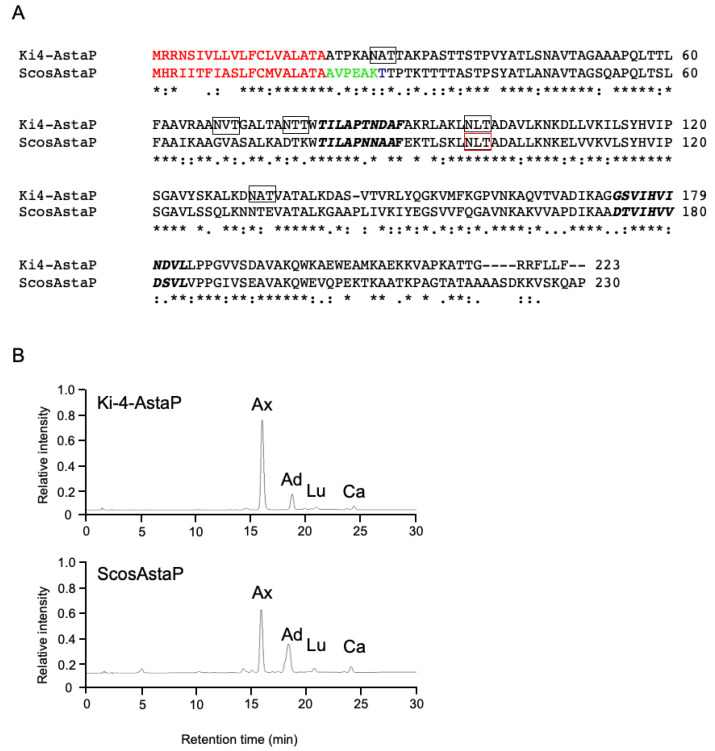
Structural comparison of Ki-4 AstaP and *S. costatus* AstaP orthologs. (**A**) comparison of the deduced amino acid sequences of Ki-4 AstaP and ScosAstaP. N-terminal signal peptides are shown in red font. The experimentally detected N-terminal sequence is highlighted in green. Potential sites for N-linked glycosylation are boxed (black: Ki-4 AstaP, red: ScosAstaP); (**B**) C_18_-HPLC elution profile of the binding carotenoids from purified Ki-4 AstaP [9] and the purified ScosAstaP. Ax, astaxanthin; Ad, adonixanthin; Lt, lutein; and Ca, canthaxanthin.

**Figure 7 marinedrugs-19-00349-f007:**
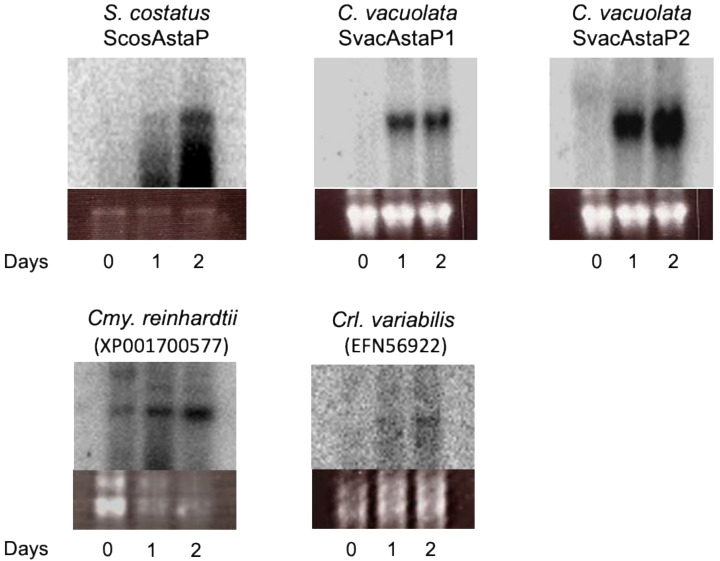
Northern blots of total RNA probed with the gene encoding AstaP orthologs from *S. costatus, C. vacuolata, Cmy. reinhardtii, and Crl. variabilis*. Strains were cultivated under photooxidative stress conditions. 0, just before the start of stress, 1–2, after one and two days of stress. Gene accession numbers are shown in parentheses. Ethidium-bromide staining of the ribosomal RNA (rRNA) to confirm the equal loading is shown below the autoradiogram.

**Figure 8 marinedrugs-19-00349-f008:**
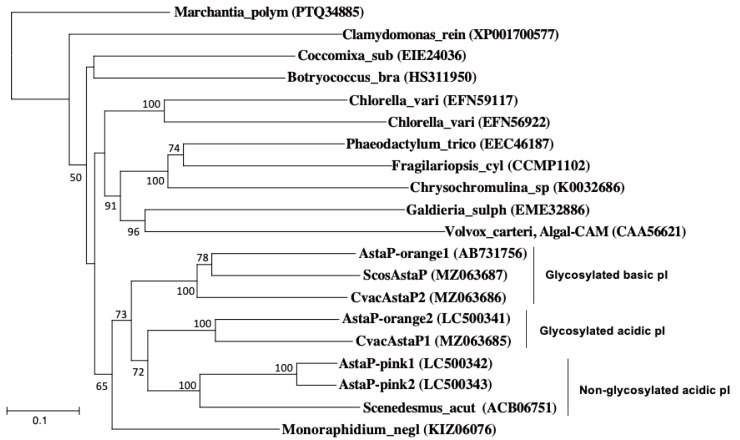
Neighbor-joining phylogenetic tree of the deduced sequence of the AstaP orthologs. A livewort homologue (Marchantia_polym: *Marchantia polymorpha*) was used as an outgroup. Chlamydomonas rein: *Chlamydomonas reinhardtii*, Chlolella_vari: *Chlorella variabilis*, Monoraphidium_negl: *Monoraphidium neglectum*, Scenedesmus_acut: *Scenedesmus acutus*, Botryococcus_bra: *Botryococcus braunii*, Coccomyxa_sub: *Coccomyxa subellipsoidea*, Fragilariopsis_cyl: *Fragilariopsis cylindrus*, Chrysochromulina sp: *Chrysochromulina tobinii*, Phaeodactylum_trico: *Phaeodactylum tricornutum*, Volvox_carteri: *Volvox carteri,* and Galdieria_sulph: *Galdieria sulphuraria*. Accession numbers for each protein are shown in parentheses. The bootstrap values >50 are indicated at the branch points.

**Table 1 marinedrugs-19-00349-t001:** Measurements of the amount of aqueous pigments after ultracentrifugation of cell extracts.

Species and Strain	Stress Conditions (with/ HL)	OD_480_ (1 g Wet Cell/10 mL)	Reference
*Coelastrella astaxanthina* Ki-4	0.7 M NaCl	1.3	[9]
*Scenedesmus obtusus* Oki-4N	0.5 M NaCl	1.0	[10]
*Coelastrella striolata* SAG 16.95	0.4 M NaCl	0.14	This study
*Coelastrella vacuolata* SAG 211-8b	0.5M NaCl	0.12	This study
*Scenedesmus costatus* SAG 46.88	0.4 M NaCl	0.16	This study
*Scenedesmus obliquus* SAG 276-3a	0.2 M NaCl	0.02	This study
*Scenedesmus obtusus* SAG 52.80	0.25 M NaCl	0.15	This study
*Chlamydomonas reinhardtii* NIES-2238	0.2 M NaCl	0.04	This study
*Chlorella variabilis* NIES-2540	0.4 M NaCl	0.33	This study

## Data Availability

The cDNA sequence data of CvacAstaP1, CvacAstaP2, and ScosAstaP have been deposited in the NCBI database under the accession numbers MZ063685–MZ063687, respectively.

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
