# Peer review of "Distribution of the Water-Soluble Astaxanthin Binding Carotenoprotein (AstaP) in Scenedesmaceae"

_marinedrugs, 2021, doi:10.3390/md19060349_

Round 1

Reviewer 1 Report

This manuscript by Toyoshima et al. performed various analyses to understand the distribution of water-soluble AstaP in the green algal family Scenedesmaceae. In general, I think the ms is clearly written and well organized. In only have a few minor comments on the ms.

Line 138. It is useful to explain why S. costatus was singled out for more detailed analyses among tested species.

Lines 194-199. The authors discussed two AstaP sequences in C. vacuolate. In the supplementary Figure 1, only sequences were listed. It would be easier for readers to understand if sequence alignment is provided.

Discussion, it is unclear why the AstaP sequences from the two model microalgae were not discussed.

Author Response

Reviewer 1

This manuscript by Toyoshima et al. performed various analyses to understand the distribution of water-soluble AstaP in the green algal family Scenedesmaceae. In general, I think the ms is clearly written and well organized. In only have a few minor comments on the ms.

Answer: Thank you for the kind comments. We have taken into consideration all your suggestions and revised the manuscript accordingly.

Line 138. It is useful to explain why S. costatus was singled out for more detailed analyses among tested species.

Answer: We thank you for the comment. We chose S. costatus in this study because its relatively high expression level of pigments (Table1). We revised a sentence (Line 140-142).

Lines 194-199. The authors discussed two AstaP sequences in C. vacuolate. In the supplementary Figure 1, only sequences were listed. It would be easier for readers to understand if sequence alignment is provided

Answer: We thank you for the comment. We added sequence alignment figure in SFig. 2.

Discussion, it is unclear why the AstaP sequences from the two model microalgae were not discussed.

Answer: We thank you for the comment. We added a sentence in Discussion (Line 241-244).

Reviewer 2 Report

This article is about the production of a novel type of proteins produced in stress conditions by different green algae. It completes the work already performed and described by the authors in a previous article. Please find bellow my comments.

Line 20 : strains and not stains.

Line 55: apparently the strains are coming from various strains collection not only one

Line 59: please define what is a non-stressed environmental condition.

Line 62: please describe the “common cultivations conditions”

Line 66: please detail the stress conditions: light cycle, temperature, pH control?, flasks volume, type of light source, inoculation density..

Line 71: can you explain what do you mean by that: “Strains Ki-4 and Oki-4N showed high tolerance to photooxidative stress”. How do you control that?

Line 78: How did you define the upper limit of salt concentrations?

Line 91: can you explain the method to concentrate the proteins?

Line 121, 126, 127, 186, fig7, fig 1: Cmy. reinhardtii and Crl. Variabilis

Line 124: no bead beating in this case?

Line 216-217: in order to obtain light stress conditions you need to consider the cell density into your culture and the mixing. That why the details of the culture conditions is very important. A very not dense culture will be much more sensitive to high light intensity than a dense culture. For Scenedesmus obliquus, the culture conditions might not have been stressful enough for it to produce the AstaP. The same thing for Chlorella variabilis. We can see in Fig 1 that it’s not stressed at all.

Author Response

Reviewer 2

This article is about the production of a novel type of proteins produced in stress conditions by different green algae. It completes the work already performed and described by the authors in a previous article. Please find bellow my comments.

Answer: Thank you for your many kind comments. We have taken into consideration all your suggestions and revised the manuscript accordingly.

Line 20: strains and not stains.

Answer: We thank you for finding a typo. We revised a word (Line 21).

Line 55: apparently the strains are coming from various strains collection not only one

Answer: We thank you for finding a typo. We revised a sentence (Line 57).

Line 59: please define what is a non-stressed environmental condition.

Answer: We thank you for the comment. We added a sentence in Materials and Methods (Line 267-269).

Line 62: please describe the “common cultivations conditions”

Answer: We thank you for the comment. We revised a sentence (Line64).

Line 66: please detail the stress conditions: light cycle, temperature, pH control?, flasks volume, type of light source, inoculation density..

Answer: We thank you for the comment. We added a sentence in Materials and Methods (Line 267-269).

Line 71: can you explain what do you mean by that: “Strains Ki-4 and Oki-4N showed high tolerance to photooxidative stress”. How do you control that?

Answer: We thank you for the comment. We revised a sentence (Line74).

Line 78: How did you define the upper limit of salt concentrations?

Answer: We thank you for the comment. As written in line 77-86, we decided the upper limit of salt concentrations at the point where the cells turned color from green to white. Then we tested the salt concentrations just below the upper limit.

Line 91: can you explain the method to concentrate the proteins?

Answer: Sorry we missed to explain. We used Amicon Ultra 0.5 mL filter (Merck, Darmstadt, Germany) to concentrate the proteins. We added this information in material and methods section (Line 283-284).

Line 121, 126, 127, 186, fig7, fig 1: Cmy. reinhardtii and Crl. Variabilis

Answer: We thank you for the comment. When we use the same abbreviation “C” for Coelastrella, Chlamydomonas, and Chlorella, it might be difficult for the readers to distinguish three genera, therefore, we used different abbreviations for each genus. We explained this abbreviation in Line 347.

Line 124: no bead beating in this case?

Answer: We thank you for suggesting our missing. We used bead beating for all the tested strains in this study. We added a sentence in Line127.

Line 216-217: in order to obtain light stress conditions you need to consider the cell density into your culture and the mixing. That why the details of the culture conditions is very important. A very not dense culture will be much more sensitive to high light intensity than a dense culture. For Scenedesmus obliquus, the culture conditions might not have been stressful enough for it to produce the AstaP. The same thing for Chlorella variabilis. We can see in Fig 1 that it’s not stressed at all.

Answer: We thank you for the comment. As we wrote in Line 267-269, we began the stress treatments for all the strains when the cells reached an OD750=1.0. In addition, we at first checked the upper limit of salt concentrations under the high light exposure conditions (Line 77-79). Then, we chose salt concentrations for each strain. Therefore, we concluded that all the strains feel photooxidative stresses, including Chlorella variabilis.

Reviewer 3 Report

This work is devoted to the study of the astaxanthin binding caroteno-2 protein (AstaP) in several strains of Scenedesmaceae. The topic is interesting. However, several comments have to be taken into consideration to improve the manuscript's quality.

  • Abstract has to be rewritten according to performed work. For example, text on lines 13-17 has to be transferred to Introduction.
  • What is “Photooxidative stress”?

As it is well known: “Photooxidative stress in plants is mostly induced by the absorption of excess excitation energy leading to over-reduction of the electron transport chains generating ROS” (REDDY A.R., RAGHAVENDRA A.S. (2006) PHOTOOXIDATIVE STRESS. In: Madhava Rao K., Raghavendra A., Janardhan Reddy K. (eds) Physiology and Molecular Biology of Stress Tolerance in Plants. Springer, Dordrecht. https://doi.org/10.1007/1-4020-4225-6_6).

In the present work, the Authors applied indeed two types of stresses to different Scenedesmaceae strains:

(1)a photooxidative stress (high light w/HL, 800 μmol photons m-2 s-1) and (2)  salt stress (different amounts of NaCl ).

So, in such a case, the Authors have to write about these two stresses and point them in the manuscript. Two different stresses can not be named as one “Photooxidative stress”.

 So, corresponding changes have to be made throughout the manuscript.

  • Why the Authors did not show the control samples: HL (high light) without salt, LL (low light) without salt, LL with salt? Please, provide the corresponding results in the main text or Supplementary.
  • Please describe in the “Material and Methods” section “normal conditions” in which cells were grown before stress application.
  • Lines 59-62. Please, indicate shortly what kind of growth medium, light intensity, temperature etc. were used in “common cultivation conditions”.
  • Figure 1. Please, describe non-stressed conditions.
  • Line 221. Please, clarify and support by data.
  • Figure 8 has to be placed in the Results section as a result of a phylogenic study.
  • Discussion has to be rewritten. Now the Authors are mainly discussing in this section Figure 8 and are doing conclusions. Please, write Discussion and write Conclusions.
  • Lines 216-217. Without showing control samples, the Authors can not say that “All the test strains, except S. obliquus, were found to express astaxanthin-binding water-soluble protein under photooxidative stress conditions”.
  • Please discuss why these two stresses together (salt and HL) induce the expression of AstaP orthologs? What hypothesis can be suggested?

Author Response

Reviewer 3

This work is devoted to the study of the astaxanthin binding caroteno-2 protein (AstaP) in several strains of Scenedesmaceae. The topic is interesting. However, several comments have to be taken into consideration to improve the manuscript's quality.

Answer: Thank you for your kind comments. We have significantly revised the manuscripts.

Abstract has to be rewritten according to performed work. For example, text on lines 13-17 has to be transferred to Introduction.

Answer: We thank you for the comment. We thought a brief introduction was needed for the abstract in this manuscript because MDPI guidelines recommended containing “background” section in the abstract.

What is “Photooxidative stress”?

As it is well known: “Photooxidative stress in plants is mostly induced by the absorption of excess excitation energy leading to over-reduction of the electron transport chains generating ROS” (REDDY A.R., RAGHAVENDRA A.S. (2006) PHOTOOXIDATIVE STRESS. In: Madhava Rao K., Raghavendra A., Janardhan Reddy K. (eds) Physiology and Molecular Biology of Stress Tolerance in Plants. Springer, Dordrecht. https://doi.org/10.1007/1-4020-4225-6_6).

In the present work, the Authors applied indeed two types of stresses to different Scenedesmaceae strains:

(1)a photooxidative stress (high light w/HL, 800 μmol photons m-2 s-1) and (2)  salt stress (different amounts of NaCl ).

So, in such a case, the Authors have to write about these two stresses and point them in the manuscript. Two different stresses can not be named as one “Photooxidative stress”.

So, corresponding changes have to be made throughout the manuscript.

Answer: We thank you for this comment. There are several reports that plants feel photooxidative stresses under abiotic stress conditions in conjunction with high light stress (Asada K, 1999, Annu. Rev. Plant Physiol, Foyer and Shigeoka, 2011, Plant Physiology) that leads to over-reduction of the electron transport chain in plant photosynthesis. We cited these reports in Introduction (Line32-34) and in Reference.

Why the Authors did not show the control samples: HL (high light) without salt, LL (low light) without salt, LL with salt? Please, provide the corresponding results in the main text or Supplementary.

Answer: We thank you for this comment. In a previous study by using the strain Ki-4 and Oki-4N, AstaP was expressed under high light conditions with salt, but not under low light condition with salt, or high light only without salt. All the algal strains showed similar growth profile to Ki-4 and Oki-4N, and they did not change cell color to orange under high light conditions (800 µmol photons m-2 s-1) without salt, or low light conditions with salt. We added a sentence about this in line 272-273.

Please describe in the “Material and Methods” section “normal conditions” in which cells were grown before stress application.

Answer: We thank you for the comment. Normal condition was previously described in our study. Briefly, tested algae were cultured in A3 medium with a 16 h/8 h regime at 26 °C under low light conditions (60 μmol photons m–2 s–1). (Line 267-269)

Lines 59-62. Please, indicate shortly what kind of growth medium, light intensity, temperature etc. were used in “common cultivation conditions”.

Answer: We thank you for the comment. Common cultivation conditions are the same conditions as normal conditions. We have rewritten this sentence to understand it easily.

Figure 1. Please, describe non-stressed conditions.

Answer: We thank you for the comment. We added the description in Figure 1 legend.

Line 221. Please, clarify and support by data.

Answer: We thank you for the comment. We estimated that S. costacus, C. vacuolata, and S. striolata were isolated under non-stressed conditions based on their isolated habitat.

Figure 8 has to be placed in the Results section as a result of a phylogenic study.

Answer: We thank you for this comment. We replaced Figure 8 and transferred the corresponding sentences from Discussion to the Results section (Line 217-220).

Discussion has to be rewritten. Now the Authors are mainly discussing in this section Figure 8 and are doing conclusions. Please, write Discussion and write Conclusions.

Answer: We thank you for the comment. We transferred a Figure 8 and the corresponding sentences to the result section (Line 217-220).

Lines 216-217. Without showing control samples, the Authors can not say that “All the test strains, except S. obliquus, were found to express astaxanthin-binding water-soluble protein under photooxidative stress conditions”.

Answer: We thank you for the comment. We put words “under the experimental conditions in this study” in the sentence. (Line 236)

Please discuss why these two stresses together (salt and HL) induce the expression of AstaP orthologs? What hypothesis can be suggested?

Answer: We thank you for the comment. There are several reports that plants feel photooxidative stresses under abiotic stress conditions in conjunction with high light stress (Asada K, 1999, Annu. Rev. Plant Physiol, Foyer and Shigeoka, 2011, Plant Physiology) that leads the over-reduction of electron transport chain. We added sentences with references in Line32-34.

Round 2

Reviewer 2 Report

Thank you for changing the manuscript according to my comments. It's clearer now.

Author Response

Reviewer 2
Thank you for changing the manuscript according to my comments. It's clearer now.
Answer: Thank you for reviewing our manuscripts. We appreciate your valuable comments.

Reviewer 3 Report

The Authors have made some corrections and adding into the text of the manuscript.

However, some concerns are still exist.

  • Discussion has to be rewritten. Conclusion part has to be placed after the discussion of own results in comparison with the already published results. Please, stress a novelty and significance of your work.
  • Please, clarify the text on lines 239-240. What do you mean when writing about the expression level of protein AstaP and habitat condition.

Author Response

Reviewer 3
Suggestions for Authors.
The Authors have made some corrections and adding into the text of the manuscript. However, some concerns are still exist.
1. Discussion has to be rewritten. Conclusion part has to be placed after the discussion of own results in comparison with the already published results. Please, stress a novelty and significance of your work.

Answer: We thank you for the comment. We have rewritten the Discussion (Line 232-261). We changed the order of the sentences, put a few new sentences highlighted in Yellow. We placed the conclusion part at the end of Discussion (Line252).

2. Please, clarify the text on lines 239-240. What do you mean when writing about the expression level of protein AstaP and habitat condition.

Answer: Thank you for your comments. We revised a sentence (Line249-251).
